# YouTube-SL-25: A Large-Scale, Open-Domain Multilingual Sign Language Parallel Corpus

**Garrett Tanzer**[*]
Google DeepMind

**Biao Zhang**
Google DeepMind

## Abstract

Even for better-studied sign languages like American Sign Language (ASL), data is the bottleneck for machine learning research. The situation is worse yet for the many other sign languages used by Deaf/Hard of Hearing communities around the world. In this paper, we present YouTube-SL-25, a large-scale, open-domain multilingual corpus of sign language videos with seemingly well-aligned captions drawn from YouTube. With >3000 hours of videos across >25 sign languages, YouTube-SL-25 is a) >3x the size of YouTube-ASL, b) the largest supervised sign language dataset to date, and c) the first or largest parallel dataset for many of its component languages. We provide baselines for sign-to-text tasks using a unified multilingual multitask model based on T5 and report scores on benchmarks across 4 sign languages. The results demonstrate that multilingual transfer benefits both higher- and lower-resource sign languages within YouTube-SL-25.[1]

## 1 Introduction

There are >300 sign languages used by Deaf/Hard of Hearing communities around the world. As minority languages, sign languages have relatively little data available and therefore—like low-resource spoken languages—are challenging to process with machine learning. The fact that sign languages are visuospatial languages represented as video with no commonly used written form creates extra challenges at every stage of development: they are more difficult to mine, filter, preprocess, and model. Datasets like YouTube-ASL [35] for American Sign Language (ASL) and BOBSL [1] for British Sign Language (BSL) make a focused effort to advance the status quo for a single sign language, but many of the world's sign languages are being left behind.

In this paper, we present YouTube-SL-25, a large-scale, open-domain corpus of multilingual sign language videos with seemingly well-aligned captions, primarily intended for translation from each sign language to its region's spoken language. By *open-domain*, we are distinguishing from datasets like AfriSign [9] and JWSign [8], which have made strides towards massive sign-multilinguality but feature a limited domain (e.g., Bible translations).

We mined these videos using a two-step process: First, like YouTube-ASL [35], we used automatic classifiers on text metadata to identify potentially relevant videos. And second—in contrast to YouTube-ASL, which paid 3 native ASL signers to filter out individual videos with wrong or misaligned captions over the course of several months, a process that isn't amenable to scaling across many sign languages—we used our own knowledge of sign languages and YouTube data to triage the videos over four days, ranking the priority of channels according to total hours of content and then auditing videos per channel, with particular attention to outliers by duration. This means that the annotations were performed with less expertise than YouTube-ASL, but in practice there are many signals that can be used to identify high-quality content even without complete understanding, as observed in work with written languages [15; 16].

---

[*]Correspondence to `gtanzer@google.com`.

[1]We release the YouTube-SL-25 video IDs under CC BY 4.0 at this link. Note that this license only applies to the video IDs and ISO 639-3 language codes, which we selected and labelled. The underlying video and caption content, as with all datasets consisting of YouTube video IDs, is subject to different licenses and should be accessed/used in accordance with the YouTube Terms of Service.

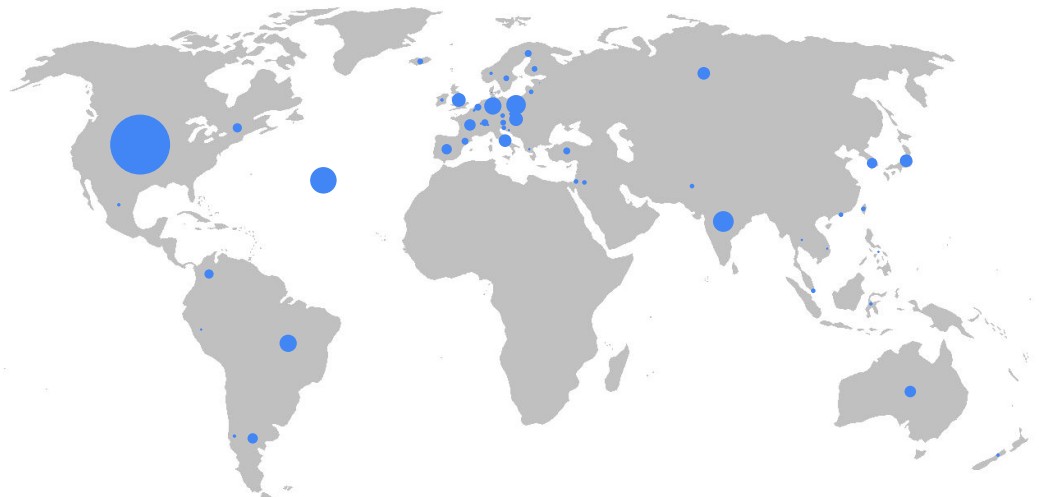

Figure 1: **World map showing the amount of content in YouTube-SL-25 for each sign language;** the area of each circle is proportional to the number of hours. The circle in the middle of the Atlantic Ocean represents International Sign. Observe that the dataset is especially lacking in representation for Central & South America, Africa, West & Central Asia, and China & Southeast Asia.

The result is a dataset with 3207 hours of videos with seemingly well-aligned captions, featuring >3000 unique signers across >25 sign languages. This is >3x the size of YouTube-ASL (984 hours), which is a subset of YouTube-SL-25, and larger than the prior largest parallel sign language dataset, JWSign (2530 hours), which is closed-domain. YouTube-SL-25 is the first or largest parallel dataset for many of its component sign languages.

Following Kreutzer et al. [15]'s call to avoid "representation washing"—i.e., overstating the number of low-resource languages supported when their dataset size is marginal—we name YouTube-SL-25 after the >25 sign languages that have at least 15 hours of representation in it. This is roughly the minimum size of datasets that have previously been released for individual sign languages, but it is still considered extremely low resource. When including the long tail, YouTube-SL-25 includes data in at least 55 sign languages.

We provide baselines for YouTube-SL-25 on sign language understanding tasks, extending the multitask mixture from FLEURS-ASL's baselines [34] to support multiple source/target languages and the sign language identification task (which is extremely understudied [6; 7; 33]). Our results on benchmarks across 4 sign languages show that both higher- and lower-resource sign languages within YouTube-SL-25 benefit from multilingual transfer.

We publicly release the YouTube-SL-25 video IDs at this link. We hope that YouTube-SL-25 can be further refined by language experts in the community and will be useful for a variety of purposes, such as general sign language pretraining and medium-quality finetuning for downstream tasks like translation, caption alignment, & sign language identification. In particular, we think that it is necessary to develop robust filtering and preprocessing tools for sign languages in order to realize the next order of magnitude increase in dataset size.

## 2   THE YOUTUBE-SL-25 CORPUS

YouTube-SL-25 is a massively multilingual corpus of sign language videos with seemingly well-aligned captions drawn from YouTube, intended primarily for pretraining sign-to-text translation models. YouTube-SL-25 is a superset of YouTube-ASL [35].

YouTube-ASL used a two-step pipeline to construct its corpus: "first, retrieval using automatic content-based annotations, and second, filtering by skilled human annotators at a per-video level" [35]. We adopt the first step with minimal modifications, but use a lower standard for the second step:

triaging/auditing by a single non-native signer to identify seemingly high-quality channels for inclusion (even without full understanding of the content in every sign language).

## 2.1 AUTOMATICALLY RETRIEVING CANDIDATE VIDEOS

We retrieved candidate videos in a similar way to YouTube-ASL [35], by fetching videos tagged with Knowledge Graph entities related to sign language generally or any individual sign language with its own tag as of July 2023.[2] As with YouTube-ASL, the recall of these tags is limited in that they are not aware of sign language in the video content itself (i.e., they do not use sign language detection and sign language identification classifiers), so they may miss videos or channels that use sign language but do not explicitly mention it. The pool of videos that we fetched from is also restricted in some generic ways, such as that the videos must be public and listed.

We applied the same filtering steps as YouTube-ASL: select only videos with manually uploaded captions, and remove videos with duration <10 seconds or >5 hours, width <480 pixels or height 360 pixels, and frame rate <15fps or >60fps. We added one additional step: remove videos where captions cover <40% of the duration.

Unlike YouTube-ASL, we did not exclude videos of conversations with more than one signer in frame at a time. While our pose-based baselines only support one person's input, we feel that there is no reason to exclude these videos from the corpus itself. We are not aware of any prior works that study sign language translation from videos with multiple signers; some datasets like the Public DGS Corpus [10] consist of conversational data, but they are recorded with one camera per signer. Being able to handle this kind of data is important both in the near term to make the most of the limited resources available and in the long term to support multi-signer applications.

The result of this process was a list of 81,623 candidate videos that might contain signed content with high-quality captions.

## 2.2 TRIAGING CANDIDATE VIDEOS WITH COARSE MANUAL ANNOTATIONS

YouTube-ASL used 3 native ASL signers as annotators to identify which of the individual candidate videos contained ASL and had high-quality, well-aligned English captions, labelling videos over the course of several months and several rounds of iteration on the annotation tool/labelling standards. This kind of annotation does not scale well to massively multilingual data because it is difficult and expensive to onboard native (or even just proficient) signers to annotate videos in tens of (extremely low resource) sign languages. Instead, we follow Kreutzer et al. [15]'s observation that many data filtering tasks can be done even without full understanding of the content. Therefore, the first author (a non-native hearing signer with experience in several sign languages, primarily ASL) served as the annotator, triaging the corpus over the course of a four-day weekend using a combination of per-channel annotations and targeted audits. The triage proceeded as follows:

First, we automatically included the YouTube-ASL videos in our corpus and removed these from the manual review. Then we grouped videos by channel and sorted by total duration in descending order, with random ordering for videos within each channel. We used heuristic classifiers based on public text metadata for each video and channel (e.g., title, description, caption language) to provisionally label each video with a sign language. For each channel, the author sampled several random videos and judged whether they matched the acceptance criteria (described below), then corrected the sign language label if relevant. If all the sampled videos were good, the channel was tentatively accepted; if the videos were bad on balance, it was rejected; if it was mixed but promising, the author investigated further and was able to exclude individual bad videos. The author triaged the first 1000 channels of 18000 (down to about 1 hour of content per channel), then continued into the 3000-6000 range, skipping videos predicted to be in ASL in order to prioritize the long tail of sign languages.[3] Finally, we sorted the videos for each channel by duration in descending order and the first author made a second pass, checking outlier videos with unusually long durations for their

---

[2]We refreshed the dataset in May 2024 to include newer videos from accepted channels, with some light manual review.

[3]An observation: the acceptance rates for videos in different sign languages were dramatically different. For example, the vast majority of content from Central & South America was live-interpreted events with small interpreters and poor caption alignment. We expect that high-quality produced content is more common in

channels to reduce heavy-hitter errors. (Channels that produce mostly high-quality, shorter content sometimes had longer live-interpreted events mixed in, which have different translation and caption quality.)

Now, a description of our annotation standard for individual videos and some shallow signals to recognize these features even without full understanding:

- The video should feature sign language. While there is such a thing as fake sign language interpreters, they do not typically create YouTube channels full of themselves gesturing nonsensically, so there is not much adversarial content to sift through. The closest thing to this is inexperienced sign language learners.[4] An annotator proficient in one sign language can generally distinguish these cases (and others like interpretive dance/gesture) even for other sign languages.

- The signer(s) should be the primary content. If a signer is interpreting spoken content, the video of the speaker (if present) should be significantly smaller than the signer. If the person is switching off between signing and speaking, or a speaker is the primary content for a significant fraction of the video, it should be rejected.

- The video should have relatively well-aligned captions. There are a few signals you can use for this: if the video has no speech track, the captions are virtually always well-aligned; these are typically channels with natively Deaf-produced content that was captioned after the fact. You can validate the alignment by seeing whether the changes in captions occur at boundaries in sign language phrases (which are marked by prosodic signals that generally do not require knowledge of the particular sign language to identify [5]). If there is a speech track, you can listen to whether the caption timing matches the speech track. We accepted videos where the captions were aligned to the speech track as long as they were not off by more than a couple of seconds; usually this corresponds to an after-the-fact voiceover translation, not live interpretation.

- Label which sign language the video is in. It was almost always possible to classify the sign language with high confidence based on the video title/description and caption language, or at worst the channel's description (including its stated country). The failures in automatic classification mostly came from International Sign, often specified as just "IS", which was not automatically tagged. In the rare cases where we could not determine the language with high confidence, we marked the sign language as unknown.

In order to sanity check the end-to-end quality of this triage process, the first author ran two additional audits on the final dataset (excluding the videos from YouTube-ASL, since we were already confident in their quality). The author rated two samples of 100 videos, one sampled uniformly per video id, and the other with videos sampled with probability proportional to their video duration. The author loaded each video and scrubbed through the frame previews across time to identify cuts where content and captioning habits may change.

In the sample with random videos by ID, we found: 1 video that strictly should not have been included in the dataset (a brief video with captioned narration), 1 borderline video with a waist-up interpreter next to a video of a speaker's face only, and 3 videos with content from learners that used appropriate vocabulary but didn't really provide a complete translation.

In the sample proportional to video duration, we found: 0.76% of content that was speech rather than signing (from a channel with otherwise entirely signed content), and 0.96% that consisted of interstitials mixed within otherwise good content, where the interstitials had signing but no captions.

We consider these error rates acceptable.

---

high-income countries, so it will be important for future work to leverage lower quality data in order to be more globally inclusive.

[4]We accept learners as long as they seem relatively competent. Models should be able to understand less proficient signers, especially given that many deaf people are not exposed to sign language in childhood and reach varying levels of proficiency [19]. Less proficient signing can be easily distinguished from native signing, so it should not harm the quality of sign language understanding (especially if the data is only used for pretraining), but more care is required for generation.

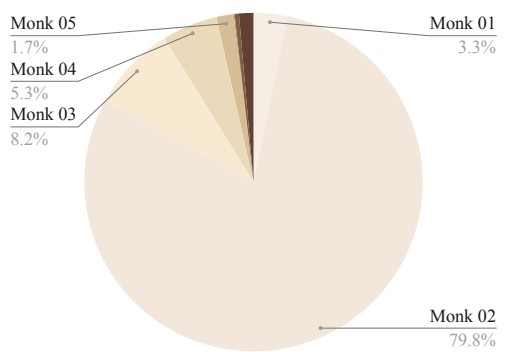 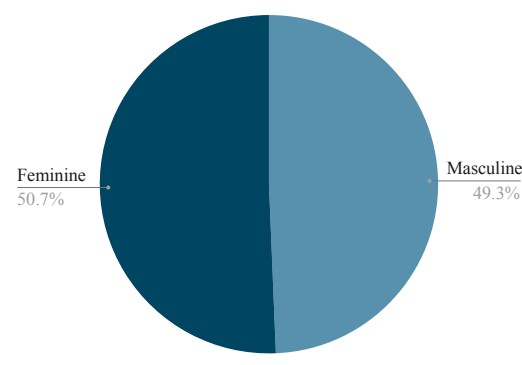

(a) **Monk skin tone ratings** [17]. Dark skin is globally underrepresented; some of this is due to the regional bias towards the Global North.

(b) **Perceived gender presentation** (nonbinary presentation not supported by the classifier). Masculine and feminine presentation are at parity.

Figure 2: **Demographic representation of YouTube-SL-25 content (proportion of hours)**, predicted with proprietary classifiers. These predictions should only be interpreted in aggregate.

## 2.3 CORPUS STATISTICS & COMPARISON TO PRIOR DATASETS

The final human-triaged YouTube-SL-25 corpus consists of 39197 videos totaling 3207 hours of content (covered by 2980 hours of captions = 2.16M captions = 104M characters)[5] across >25 sign languages. Like in YouTube-ASL, we approximately lower bound the number of unique signers using the number of unique channels, 3072, and expect that this is quite an underestimate as channels (or even individual videos) featuring multiple signers are common. This is only somewhat larger than YouTube-ASL's 2519 channels, which is the main downside of our triage-based annotation approach: it captures the bulk of the hours but not the long tail of unique channels.

As discussed in Uthus et al. [35]'s related work section, to the best of our knowledge YouTube-ASL and now YouTube-SL-25 are the only sign language translation datasets that aggregate content from a wide variety of automatically mined channels, as opposed to others which either pay participants to create new data [4; 36; 14; 10] or curate content from a small number of content creators [2; 32; 1; 31; 21; 8; 12; 18]. This has implications for diversity in topics/content and video production style. Note that even though YouTube-SL-25 is open-domain overall, there may be discretization effects in the long tail of languages, where a language is represented by only a few channels and therefore in effect is closed-domain.

See Figure 1 for a depiction of the relative weight of each language on a world map and Table 1 for a breakdown of hours by language, comparing to the largest prior parallel dataset for each language (both open- and closed-domain). Note that prior works have studied sign languages with no representation in YouTube-SL-25, like AfriSign [9] & JWSign [8] (primarily languages in Central & South America and Africa), and CSL-Daily [40] (China); these languages aren't included in the table. The distribution of language data can be attributed to population size, economic development, and political factors. Other supplementary sources of data, such as Bible translations [9; 8] and national interpreted broadcasts [1; 3], could provide more balanced data distributions.

See Figure 2 for an estimated demographic breakdown of YouTube-SL-25 by skin tone and perceived gender presentation, predicted by automated classifiers. Dark skin is globally underrepresented in the dataset, with only 1.9% (60 hours) of the data at Monk 6-10. Some of this can be explained by overrepresentation of countries in the Global North (and the only two countries in the Global South with a significant fraction of the dataset, Brazil and India, have substantial variation in skin tone). Ensuring demographic fairness remains an important problem for future work.

---

[5]We used classifiers to evaluate whether the captions were toxic, hateful, violent, and sexual with confidence >0.8, which flagged <1% of captions. We manually audited the captions flagged with highest confidence and none merited exclusion from the dataset: the "toxic" and "hateful" content were false positives, the "violent" content was news about violent events, and the "sexual" content was news or educational content about sex.

Table 1: **Hours of content in YouTube-SL-25 across sign languages, compared to the largest prior publicly available parallel dataset for each language (both open- and closed-domain).** Some of these sign languages have more commonly used names, like "Libras" for "Brazilian SL", but we refer by region here for convenience. Note that International Sign (ils) is a pidgin that arises at international Deaf conferences, in contrast to all the other sign languages on this list, which are complete natural languages. With the significant exception of YouTube-ASL, most prior datasets consist of mutually exclusive data that could supplement YouTube-SL-25.

| sign language | iso 639 | #videos | #channels | #hours | largest prior (open) | #hours | largest prior | #hours |
|---|---|---|---|---|---|---|---|---|
| American | ase | 16724 | 2523 | **1394** | YouTube-ASL [35] | 984 | " | " |
| International | ils | 1634 | 14 | **285** | - | - | - | - |
| Indian | ins | 3023 | 8 | 209 | iSign [13] | **252** | " | " |
| Polish | pso | 1698 | 34 | **137** | - | - | JWSign | 61 |
| German | gsg | 1024 | 65 | **108** | DGS Corpus [10] | 50 | " | " |
| Brazilian | bzs | 846 | 30 | 101 | - | - | JWSign | **211** |
| British | bfi | 1026 | 60 | 74 | BOBSL [1] | **1660** | " | " |
| Hungarian | hsh | 1687 | 9 | **70** | - | - | JWSign | 16 |
| Australian | asf | 1098 | 25 | **67** | Auslan-Daily [31] | 45 | " | " |
| Italian | ise | 929 | 21 | 63 | - | - | JWSign | **115** |
| Japanese | jsl | 1075 | 21 | 62 | - | - | JWSign | **67** |
| Russian | rsl | 715 | 9 | 60 | - | - | JWSign | **111** |
| French | fsl | 900 | 23 | 49 | Mediapi-RGB [21] | **86** | " | " |
| Korean | kvk | 325 | 10 | 38 | - | - | JWSign | **93** |
| Colombian | csn | 212 | 4 | 37 | - | - | JWSign | **128** |
| Spanish | ssp | 701 | 13 | 36 | - | - | JWSign | **85** |
| Dutch | dse | 450 | 14 | **35** | - | - | JWSign | 1 |
| Argentine | aed | 160 | 9 | 34 | - | - | JWSign | **107** |
| Quebec | fcs | 128 | 7 | **26** | - | - | JWSign | 20 |
| Catalan | csc | 239 | 7 | **26** | - | - | - | - |
| Pakistani | pks | 581 | 1 | **25** | - | - | - | - |
| Swedish | swl | 410 | 23 | **22** | - | - | JWSign | 14 |
| Turkish | tsm | 189 | 8 | 18 | E-TSL [41] | **24** | " | " |
| Swiss German | sgg | 185 | 6 | 18 | SRF23 [18] | **437** | " | " |
| Israeli | isr | 345 | 14 | **17** | - | - | JWSign | 1 |
| Fenno-Swedish | fss | 174 | 8 | **16** | - | - | - | - |
| Finnish | fse | 196 | 13 | 14 | Corpus FinSL [28] | 15 | JWSign | **35** |
| Austrian | asq | 177 | 11 | **13** | - | - | JWSign | 1 |
| Taiwan | tss | 130 | 5 | 13 | - | - | JWSign | **14** |
| Czech | cse | 185 | 11 | 12 | - | - | JWSign | **18** |
| Icelandic | icl | 214 | 6 | **11** | - | - | - | - |
| Irish | isg | 34 | 2 | **10** | - | - | JWSign | 2 |
| Slovenian | – | 139 | 6 | **10** | - | - | JWSign | 1 |
| Indonesian | inl | 126 | 2 | 9 | - | - | JWSign | **30** |
| Jordanian | jos | 132 | 2 | **9** | - | - | - | - |
| Norwegian | nsl | 82 | 9 | **8** | - | - | JWSign | 1 |
| Singapore | sls | 102 | 5 | **8** | - | - | JWSign | <1 |
| Thai | tsq | 106 | 1 | 7 | - | - | JWSign | **8** |
| Hong Kong | hks | 117 | 9 | 6 | TVB-HKSL [20] | **14** | " | " |
| Lithuanian | lls | 160 | 6 | **6** | - | - | - | - |
| Slovakian | svk | 59 | 4 | 4 | - | - | JWSign | **14** |
| Chilean | csg | 74 | 5 | 4 | - | - | JWSign | **100** |
| New Zealand | nzs | 79 | 8 | **4** | - | - | JWSign | 2 |
| Croatian | csq | 24 | 2 | 4 | - | - | JWSign | **5** |
| Mexican | mfs | 53 | 3 | 4 | - | - | JWSign | **184** |
| Philippines | psp | 26 | 2 | 3 | - | - | JWSign | **21** |
| Vietnamese | – | 36 | 1 | 3 | - | - | JWSign | 3 |
| Peruvian | prl | 71 | 4 | 2 | - | - | JWSign | **81** |
| Flemish | vgt | 57 | 4 | 2 | VGT-RAW [3] | **100** | " | " |
| Swiss French | ssr | 33 | 2 | **2** | Signsuisse [18] | 13 | - | - |
| Greek | gss | 22 | 1 | 2 | Elementary23 [36] | **71** | " | " |
| French Belgian | sfb | 25 | 5 | 2 | - | - | JWSign | **3** |
| Swiss Italian | slf | 30 | 2 | 1 | SwissSLi [11] | **10** | - | - |
| Danish | dsl | 21 | 5 | 1 | - | - | JWSign | 1 |
| Estonian | eso | 12 | 6 | 1 | - | - | JWSign | 1 |

## 3 BASELINES

We demonstrate the value of YouTube-SL-25 with baselines for sign-to-text translation and sign language identification across benchmarks for 4 sign languages.

### 3.1 SETUP

We build on the unified modeling approach adopted by the FLEURS-ASL baseline mixture [34], which are an evolution of the YouTube-ASL baselines [35]. We adopt this approach as a convenient vehicle for our data ablations and to illustrate the range of tasks that can be constructed from the YouTube-SL-25 data; the particular details are not core to this work. In brief, the approach finetunes a pretrained encoder-decoder language model (T5-v1.1 Small) to take as input 256 tokens of text (control tokens and caption context) and 512 frames of MediaPipe Holistic landmarks at half frame rate (corresponding to a random video span), and output 256 tokens of text (including timestamps). The training examples are randomly sampled 34-second clips from the YouTube-SL-25 source videos, where the captions preceding and following the clip may be used as context to mitigate caption misalignment.

The only modification we make is to extend the mixture to support multiple source and target languages, as well as the sign language identification task with probability 0.1 (where it classifies the language and then proceeds to translate the input). Like in FLEURS-ASL, we train first on caption-level clips and then an equal mixture of caption-level and random clip-level data. See Figure 3 for a depiction of the modifications to the control token format.[6]

We compare three models: T5 with only text pretraining (i.e., the original pretrained T5 Small), T5 with continued training on YouTube-SL-25's 1400-hour ASL subset, and T5 with continued training on the full YouTube-SL-25 dataset. See Appendix A for training details.

### 3.2 COMPARISON TO PRIOR METHODS

See FLEURS-ASL [34] for discussion of how their baseline approach (in particular, training on random clips and caption tracks rather than tightly clipped individual captions) relates to prior modeling work. Insofar as YouTube-SL-25 is slightly noisier than YouTube-ASL, we slightly push the boundaries in terms of training on weakly aligned data. With respect to our modifications: modeling multiple sign languages in one model is not new [38; 39; 9; 8]. We are not aware of any prior works that train sign language identification as a task within translation models (the few prior works on sign language identification [6; 7] use low-level phonetic features only), but this is not especially novel given its use in Whisper for speech input [25].

### 3.3 RESULTS

We provide sentence-level translation (scored with BLEURT [29]) (specifically, BLEURT-20 [24]) and sign language identification (scored with top-1 accuracy) results on benchmarks for 4 sign languages: American Sign Language (ase, FLEURS-ASL [34] & How2Sign [4]), Swiss German Sign Language (sgg, WMT23 SignSuisse [18]), Swiss French Sign Language (ssr, WMT23 SignSuisse [18]), and Swiss Italian Sign Language (slf, WMT23 SignSuisse [18]). We provide both zero-shot scores and finetuned scores where relevant.[7]

Table 2 shows quantitative results on both tasks, and Table 3 shows a sample of qualitative translation examples. We see that on both translation and lang id, sign language pretraining is substantially better than none (as in Uthus et al. [35]) and multilingual transfer helps both the higher-resource

---

[6]We tried to change the pretrained model from T5-v1.1 Small [26] to mT5 Small [37] so that languages besides English could benefit from pretraining and better tokenization, but in initial experiments mT5 took about 1/3 more steps to converge and achieved worse results. We therefore ran the full set of experiments with T5 only.

[7]For translation, the model is separately finetuned for each dataset, checkpoint selected based on BLEU on the validation set. For sign language identification, zero-shot scores mean that the model is briefly finetuned on YouTube-SL-25 rebalanced to the 4 sign languages with equal weight, and finetuned scores mean that the model is finetuned on an equally weighted mixture of the benchmarks' training sets. We don't finetune on FLEURS-ASL, so the finetuned langid scores are after finetuning on How2Sign.

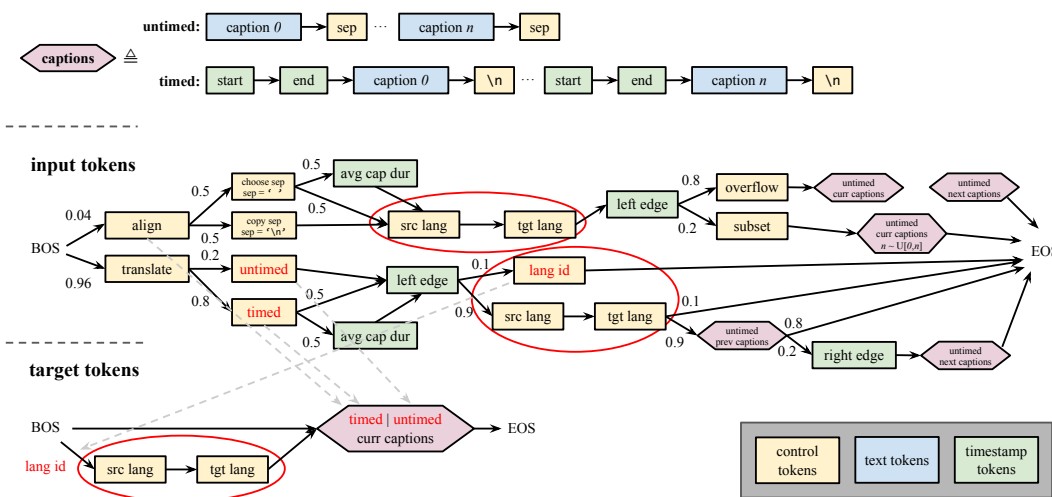

Figure 3: **Unified document-level sign-to-text training, extended for multilinguality;** modified from Figure 2 of Tanzer [34]. New additions circled in red. *At a high level:* The mixture allows training on random crops from videos with timed caption tracks (without strict sentence-level alignment). It supports multiple tasks such as translation and caption alignment using timestamps represented as text, and captions surrounding the given clip as context. We straightforwardly extend the task format from monolingual to multilingual. *At a low level:* For caption alignment, source and target language are provided unconditionally. For translation, source and target language are provided with probability 0.9 and predicted with probability 0.1. When the sign language identification task is active, we avoid conditioning on the target spoken language (either with an explicit language code, or by providing text context in the input) because it makes the task easier or trivial. (Most naturally occurring data pairs one sign language with one spoken language.)

sign language (ASL) and the lower-resource sign languages within YouTube-SL-25. Some prior works [9; 8] did not see strict improvement from modeling additional sign languages; we expect that this was a model capacity issue (during pretraining, or from finetuning on multiple languages simultaneously rather than one at a time). As expected, finetuning gives large gains—especially when the language is poorly represented in YouTube-SL-25—and benefits from multilingual pretraining. For example, our finetuned sgg result of 37.7 BLEURT (7.5 BLEU) far exceeds the top WMT23 score of 23.6 BLEURT (0.3 BLEU).

Note that for sign language identification, finetuning is somewhat ill-conceived because some datasets have train/test signer overlap or just generally unique recording conditions that could be used a shortcut. For example, langid quality on FLEURS-ASL decreases when pretrained on ASL data and finetuned on How2Sign, even compared to no sign language pretraining—with accuracy varying dramatically across the 3 signers within the test set, averaging 18.3% vs. 94.9% vs. 77.0%. This suggests that different kinds of pretraining may cause different features to be surfaced during finetuning, which may overfit or not transfer with respect to broader domains. Our zero-shot results are nontrivial despite the severe class imbalance. A proper sign language identification benchmark needs more consistency across languages; we were unable to use SP-10 [38] for licensing reasons.

## 4 LIMITATIONS

As discussed in Section 2, each step of the corpus curation comes with limitations: the automatic tagging step misses content in sign language that does not mention sign language (in the video itself or in metadata) (§ 2.1), the manual triage step trades off quality and small-channel exhaustiveness for annotation effort (§ 2.2), and the result has issues with representativeness—the distribution of languages and skin tone in particular (§ 2.3). The dataset is also small overall in comparison to MT datasets for spoken languages; we expect that more data will be needed to reach usable translation quality in generality, but this may be sufficient for more narrowly scoped tasks. Because YouTube-SL-25 (like YouTube-ASL) has so much signer variety (appearance, recording environment, proficiency,

Table 2: **Baseline results for sentence-level translation and language identification on benchmarks for 4 sign languages.** Translation is measured with BLEURT, lang id with top-1 accuracy: we report zero-shot / finetuned scores where relevant. We compare the results of no sign language pretraining vs. pretraining on the ASL subset of YouTube-SL-25 vs. the full YouTube-SL-25 dataset.

| | ase | | sgg | ssr | slf |
|---|---|---|---|---|---|
| pretrain set | FLEURS-ASL | How2Sign | | WMT23 SignSuisse | |
| *translation* | | | | | |
| None | - / - | - / 31.2 | - / 26.2 | - / 7.2 | - / 22.8 |
| YT-SL-25 (ASL) | 33.7 / - | 30.2 / 46.6 | 9.0 / 27.9 | 5.4 / 11.1 | 9.3 / 20.8 |
| YT-SL-25 (Full) | 40.1 / - | 29.7 / **47.9** | 9.5 / 37.7 | 6.8 / **18.8** | 9.3 / **25.2** |
| *lang id* | | | | | |
| None | - / 98.4 | - / 99.6 | - / 19.6 | - / 98.8 | - / 87.6 |
| YT-SL-25 (ASL) | 100.0 / 66.7 | 89.7 / 99.9 | 54.0 / 89.6 | 6.0 / **100.0** | 1.6 / **99.6** |
| YT-SL-25 (Full) | 100.0 / 99.9 | 92.7 / **100.0** | 72.0 / **99.6** | 0.4 / **100.0** | 64.8 / **99.6** |

Table 3: **Qualitative examples for sentence-level translation across datasets.** Examples selected randomly without cherrypicking (from examples already highlighted in previous papers if available).

| language | setting | text |
|---|---|---|
| ase | *reference* | And that's a great vital point technique for women's self defense. |
| | *prediction* | It's a great point for women to self-protection. |
| sgg | *reference* | Dieses Buch enthält ein Vorwort, welches sich Präambel nennt. |
| | | *(This book contains a foreword called a preamble.)* |
| | *prediction* | Dieses Buch hat eine Wörtchenschilde. |
| | | *(This book has a word shield.)* |
| ssr | *reference* | Le fiancé de mon amie va bientôt s'engager dans l'armée. |
| | | *(My friend's fiancée is going to join the army soon.)* |
| | *prediction* | Mon amie est à l'armée. |
| | | *(My friend is in the army.)* |
| slf | *reference* | Lei è allergica alle punture d'api. |
| | | *(She is allergic to bee stings.)* |
| | *prediction* | Lei ha un'ape migliore. |
| | | *(She has a better bee.)* |

and now also number of signers in frame at a time), it may be difficult to use it to train consistent sign language generation models.

Our baselines are limited in that we just use FLEURS-ASL's multitask training mixture as a way to pretrain on slightly noisy data and don't engage with the discourse-level or timestamp tasks it enables. It is difficult to evaluate these tasks (plus sign language identification) on a collection of disparate benchmarks that primarily focus on sentence-level translation; this is on top of the fact that there are not enough benchmarks (especially ones with suitable licensing) to evaluate the vast majority of sign languages in YouTube-SL-25. This underscores the importance of extending FLEURS-ASL to even more sign languages.

## 5 CONCLUSION

In this paper, we presented YouTube-SL-25, a multilingual, open-domain corpus of sign language videos with seemingly well-aligned captions, with >3000 hours of content across >25 sign languages. We achieved this efficiently without hiring tens of language-specific annotators by performing a triage grouped by channel and sorted by duration of content; this comes at the expense of the long tail of unique signers and some quality assurance. We demonstrated the value of YouTube-SL-25 with experiments in sentence-level translation and sign language identification which demonstrate multilingual transfer benefits both higher- and lower-resource sign languages. We hope that YouTube-

SL-25 and our account of how we curated it will serve as a foundation for research towards the ultimate goal of making technology inclusive for Deaf/Hard of Hearing signers worldwide.

## ETHICS STATEMENT

The ethical considerations of this work are very similar to YouTube-ASL. We release publicly available YouTube videos only as video IDs so that deletions are automatically reflected in the corpus. We train our baseline models on MediaPipe Holistic skeletons as a form of anonymization. Another work that uses YouTube-ASL, Rust et al. [27], explores an orthogonal approach for anonymization in direct video modeling: pretraining with the face blurred and finetuning with it unblurred on separate data. The field should continue to study this topic to develop a better understanding of the tradeoffs of different approaches for responsible use of the data.

While we provide estimated demographic breakdowns of the signers in YouTube-SL-25 along some dimensions, even if the dataset were globally representative, this would not guarantee that models trained on it would have equal performance across these attributes or others. And this intersects with sign multilinguality: models may perform worse for some demographics only in some sign languages due to cross-sectional effects. For example, sign language identification may be biased by the signer's race as a proxy for nationality (itself a proxy for language) due to correlations in the training distribution—to the extent that race can be inferred from the input representation. It is important to evaluate models in the context of their specific intended use cases to ensure that they robustly deliver on being useful for Deaf/Hard of Hearing users and avoid the trap of overpromising/underdelivering that has plagued sign language technology historically.

## ACKNOWLEDGEMENTS

We thank Manfred Georg and Caroline Pantofaru for institutional support; Anelia Angelova, Chris Dyer, and anonymous reviewers for feedback on drafts of this paper; David Uthus for general help with infrastructure; and Krishna Somandepalli, Katie Zhang, & Simon Wang for assistance on the fairness analysis.

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

# A    TRAINING DETAILS

We train all of our models with Adafactor [30] with base learning rate 0.001.

We pretrained our two T5v1.1 Small models (on Youtube-SL-25's ASL and Full sets) on 64 TPUv3s for 210k and 430k steps respectively (switching from pure caption-level training to 1:1 caption-level:random clip-level training once the model appeared to have converged, then stopping again after re-convergence, both according to BLEU on the How2Sign val set, like in FLEURS-ASL [34]). Each 1k steps took about 8 minutes to train. We also pretrained an mT5 Small model for about 600k steps, which was underperforming so we didn't run the complete set of experiments for it.

We finetuned the sentence-level translation models on 16 TPUv3s with a batch size of 32 until convergence; this took about 10k steps for WMT23 SS DSGS and at most 2.5k steps for the other datasets.

We finetuned the language identification models on a mixture of data for the four languages with equal weight, either subsets of YouTube-SL-25 (for zero-shot results) or the training splits from the downstream benchmarks (for finetuned results). We used 16 TPUv3s with a batch size of 32 until convergence, with up to 3k steps.

# B    BLEU SCORES

See Table 4 for sentence-level translation BLEU [22] scores, rather than BLEURT from Table 2 in the body of the paper. We use sacreBLEU [23] with `intl` tokenizer.

Table 4: **Baseline results for sentence-level translation on benchmarks for 4 sign languages**, measured in BLEU (zero-shot / finetuned).

| | ase | | sgg | ssr | slf |
|---|---|---|---|---|---|
| pretrain set | FLEURS-ASL | How2Sign | | WMT23 SignSuisse | |
| None | - / - | - / 2.6 | - / 2.2 | - / 1.3 | - / 1.1 |
| YT-SL-25 (ASL) | 3.4 / - | 3.6 / 14.5 | 0.1 / 5.4 | 0.4 / 5.9 | 0.1 / 4.0 |
| YT-SL-25 (Full) | 4.4 / - | 4.2 / **15.4** | 0.5 / **7.5** | 1.1 / **7.0** | 0.1 / **5.2** |

