# OpenReview forum: "YouTube-SL-25: A Large-Scale, Open-Domain Multilingual Sign Language Parallel Corpus"
_ICLR.cc/2025/Conference — ICLR 2025 Poster_

### Official Review · Reviewer_Yfwn · 2024-10-21

**Soundness:** 3
**Presentation:** 2
**Contribution:** 4
**Rating:** 8
**Confidence:** 4

**Summary:**

The authors present YouTube-SL-25: a new dataset for sign language machine translation. This dataset is the first of its kind, in that it is significantly larger than predecessors and includes many languages for which previously, no labeled sign language machine translation data was available.

The dataset contains more than 3000 hours of video data for more than 25 sign languages, and was collected in two steps. First, YouTube was mined based on video tags related to sign language. Second, the authors (non-signers) performed a manual audit of the videos to identify high-quality video channels. Videos from those channels were used. An overview of the included sign languages, and relation to previous datasets for these sign languages (if any) is given in Table 1.

Key features of the dataset are as follows: it is open-domain and it includes videos with multiple signers in the frame. These features make it an important dataset for future research that will eventually lead to sign language machine translation systems that are usable in the real world.

A baseline method, adapted from previous work, is also presented. It is evaluated for four sign languages included in YouTube-SL-25. This method shows that multilingual training benefits both low-resource and (surprisingly) high-resource sign languages.

**Strengths:**

This manuscript presents a valuable new dataset for the sign language machine translation community. The dataset is large and multilingual, and opens up new possibilities for sign language machine translation research in many previously underrepresented sign languages. As such, this manuscript is a valuable contribution to the field of sign language processing.

# Originality

This manuscript presents a labeled multilingual corpus for sign language translation. Previous corpora for sign language machine translation have often been monolingual (including the YouTube-ASL dataset, on which this manuscript builds). Research (in linguistics and in sign language processing) shows that sign languages have common elements, and therefore multilingual corpora can be beneficial for discovering and exploiting these common elements. The dataset is the first of its kind for several of the included low-resource sign languages and enables sign language translation research for these languages.

# Quality

The authors perform a partial analysis into the demographics of the collected data, in terms of skin tone and perceived gender.

# Clarity

The sections of the manuscript that discuss the dataset are clearly written and easy to follow. This is in contrast to the section on baseline methods, which is harder to follow (see "Weaknesses").

# Significance

A major problem in the domain of sign language translation is a lack of labeled data. The authors present a new dataset of unprecedented scale, with data for more than 25 languages and more than 3000 hours. Such resources are highly sought after and valued by the sign language processing community.

**Weaknesses:**

# Writing style and presentation

The authors make excessive use of footnotes (11 footnotes on 8 pages), which makes the text difficult to follow. I suggest moving some footnotes to the main text (3, 4, 6, 8, and 11 can definitely be in main text).

Figure 3 is quite complex and is nearly impossible to understand without first thoroughly reading the previous work from which it was adapted ([32]). I suggest adding an explanation in the figure caption, or, if constrained by the page limits, in appendix. This way, readers do not have to read and understand [32] to continue reading this manuscript past Figure 3.

# Methodology

The auditing of channels was performed by only one non-native hearing signer. The authors claim that this person is "experienced" in several sign languages, but they do not clarify what this experience entails and how this experience makes them a suited candidate for this auditing process. This leads to doubts about the robustness of the auditing process. Could the authors specify what "experience" means in this case? Having at least three auditors would also lead to a more robust auditing process.

The baseline method that is presented appears to be rather complex, especially compared to simple but effective sign language machine translation methods (e.g., https://arxiv.org/abs/2003.13830). It would be to the benefit of the reader if the method was simple, such that results are easier to interpret (the same argument is made by [33], which is cited by the manuscript's authors). Simplifying the baseline method would also give more room for analyzing results on more than just four sign languages and for further analyses of the results. Perhaps the authors could explain in the text why such a complex baseline method was chosen.

The results of the baseline method are discussed in limited detail. Scores are only provided in one translation metric (two, if including the appendix), and for the four evaluated languages, only one translation example is given per language. It would be interesting to see at least one good and one bad translation per language, and more than one if possible.

# Relation to previous work

The authors compare their dataset primarily to two previously released datasets:
- YouTube-ASL, because the same approach to data collection was used
- JWSign, because it is also a multilingual sign language machine translation dataset

The authors do not mention RWTH-PHOENIX-Weather 2014 T (https://www-i6.informatik.rwth-aachen.de/~koller/RWTH-PHOENIX-2014-T/) at all, despite it being a pivotal historical contribution in the field. Despite the limitations of PHOENIX 2014T (see, e.g., https://aclanthology.org/2022.wmt-1.71.pdf), I believe it is important to at least mention the dataset in passing to place this manuscript in the broader context of sign language machine translation.

The baseline method is adapted from [32], but there is little comparison to previous work in the field. The manuscript, in particular section 3, would benefit from an explanation as to why this method and these input features (MediaPipe) are used. If space permits, the authors should present a short section on related work in the design of sign language machine translation. If space is limited, they can refer to recent survey papers.

# Quality

On page 4, the authors mention "whether the changes in captions occur at boundaries in sign language phrases (which are marked pretty universally with pauses and nonmanual signals like head tilt or eyebrow movement)". It is my understanding that the boundaries of sign language phrases are a contentious topic in sign language linguistics, and that there certainly is no consensus on signals that are universal with respect or multiple sign languages. The authors should include a citation for this claim, or do not express this claim so strongly.

**Questions:**

# Figures

- Figure 3: Please clarify the meaning of the acronym "w.p." in the figure caption.

# Main text

- Page 2: Please provide exact figures for the amount of unique signers and sign languages, instead of >3000 and >25.
- Page 3: Can you clarify in what way the first author is "experienced" with multiple sign languages?
- Page 4: You mention in footnote 5: "We accept learners as long as they seem relatively competent". How was their competence evaluated? Is this competence evaluated by just one signer (the manuscript's first author), and how significant is this evaluation?
- Page 5: Please add a reference [15] for the Monk skin tone ratings to the main text (final paragraph) as well, not just the figure caption.

---

> ### Author Response · Authors · 2024-11-22
>
> Thank you for your valuable comments!
>
>
> **Re Weaknesses:**
>
>
> *Writing style and presentation:*
>
> Thanks for your suggestion! We will move some footnotes into the body of the paper in the revised version. We will also add more clarification of Figure 3.
>
>
> *Methodology:*
>
> We will elaborate more on the annotator’s sign language background in revisions; we didn’t go into too much detail to avoid anonymity concerns. In brief: formal instruction in some (primarily ASL with ongoing usage), self-study in others. While more annotators can always be beneficial, the annotations to produce the dataset were relatively coarse and permissive (both in terms of the blanket effect on channels, and the criteria for inclusion in the dataset). The salient potential bias in the annotation procedure, as we acknowledge in Limitations, is that it reduces the number of unique signers because it favors large channels.
>
> We chose to build off the FLEURS-ASL baselines because their random crop-based training made sense for YouTube-SL-25’s videos, which we expect to be slightly more misaligned than even YouTube-ASL. You are right that the additional tasks in the mixture were less important, but we wanted to show how you could straightforwardly extend such a multitask model to support additional language pairs and sign language identification. We briefly reexplain the task format for context but don’t view the details as especially important. This method is still “simple” in the sense of the YouTube-ASL baselines (forgoing inductive bias that won’t matter with scale), and the ablations are all about demonstrating ability to leverage data scale.
>
> Are there particular metrics you would like to see besides BLEU and BLEURT? We picked the best classical metric and the best neural metric commonly used by other sign language and machine translation works.
>
>
> *Relation to previous datasets:*
>
>  We initially focused on datasets that were the largest for each language, but we will add a reference to PHOENIX on line 248 (after “curate content from a small number of content creators”) in recognition of its historical significance.
>
>
> *Relation to previous modeling techniques:*
>
> We will add some motivation for MediaPipe Holistic in revisions: it is a form of anonymization (which is desirable when training on mined web data with human appearance), and it reduces the complexity/potential bias in handling diverse visual inputs because it was trained and evaluated independently for fairness. (Of course, this comes at the expense of introducing a frozen bottleneck.) But as mentioned above we do not view the details of the baseline setup as being key decision points; they are just inherited from reasonable prior work to host the data ablations.
>
>
> *Quality:*
>
> We will replace the parenthetical with “boundaries in sign language phrases (which can often be identified based on pauses and nonmanual signals [cite])” with a reference to “Seeing sentence boundaries” by Fenlon et al. Qualitatively, when there is enough space between signed sentences it is easy to see whether the captions are well-aligned—and alignment tends to remain consistent across a video/channel, so you only need to find a few easy points to check.
>
>
>
> **Re Questions:**
>
> *Figures:*
>
> * We can expand “w.p.” to “with probability”.
>
> *Main text:*
>
> * We do not have an exact number of unique signers, just a lower bound based on the number of unique channels (described on line 240), so we think mentioning a specific number here would be unhelpful. We mention the specific number of languages (55) in the next paragraph (lines 82-87), contextualized with the fact that most of these languages are very poorly represented.
> * We will add detail about the first author's sign language experience.
> * Learner competence: This was indeed evaluated by the one annotator based on heuristics, relatively permissively. For the dataset’s primary use case of sign language translation (=understanding), it is not especially harmful to include learners since (to the extent that they sign in a distinguishable way) it just means models will be able to understand people with poor signing proficiency. As we mention on lines 459-462, the wide variety in YouTube-SL-25 poses additional challenges for sign language generation (you want to train on diverse data but generate a particular kind of signing), but not in a way that is unique with respect to learners or that would merit exclusion from the dataset.

---

> > ### Comment · Reviewer_Yfwn · 2024-11-26
> >
> > Thank you for your responses to my remarks and questions. The additional information and proposed minor revisions to the paper should be sufficient to address all my concerns about the paper.

---

### Official Review · Reviewer_nLUa · 2024-11-02

**Soundness:** 3
**Presentation:** 3
**Contribution:** 2
**Rating:** 6
**Confidence:** 4

**Summary:**

The paper introduces YouTube-SL-25, a multilingual corpus of links to sign language videos. This dataset is designed to support pretraining for sign language models, and fine-tuning for various downstream sign language tasks. The paper provides benchmarks of sign-to-text translation and sign language identification across 4 sign languages.

**Strengths:**

(1) The paper is well-organized and easy to follow.

(2) YouTube-SL-25 addresses the shortage of data for sign language research. This large-scale dataset serves as a valuable resource for tasks like translation, caption alignment, and sign identification.

(3) Models pretrained on YouTube-SL-25, compared to YouTube-ASL, show improvements across various sign languages, highlighting the effectiveness of this multilingual dataset.

**Weaknesses:**

(1) While the dataset offers significant benefits, much of the technical training relies on the FLEURS-ASL baseline [1]. Clarifying the unique technical contributions of this paper would enhance its value.

(2) The dataset’s value is dependent on the availability of the videos. If these videos were to be removed, the overall impact of the paper would be significantly limited. To mitigate this risk, open-sourcing the framework used to filter sign language videos would benefit researchers in the long run and facilitate further development.

(3) Currently, there is no information on whether the pretrained models will be open-sourced. Providing access to these models would greatly benefit the research community, allowing other researchers to leverage large-scale pretrained models and explore new approaches in sign language tasks. This is especially important as links to the videos may become unavailable over time.

-----

[1] Garrett Tanzer. Fleurs-asl: Including american sign language in massively multilingual multitask evaluation, 2024.

**Questions:**

Please see weaknesses for suggestions and questions.

---

> ### Author Response · Authors · 2024-11-22
>
> Thank you for your valuable comments!
>
> Re Weaknesses:
>
> 1. We believe this is a misunderstanding. The contributions of this paper are primarily with respect to the dataset (the final artifact) and new method of constructing it (which uses less/less skilled human annotation). Secondarily, our baselines contribute some new kinds of results (e.g. multilingual transfer at scale, sign language identification). We do not claim particular modeling contributions. For this reason we selected the primary area “datasets and benchmarks”.
>
> 2+3. Unfortunately we are currently unable to open source anything beyond the dataset, either because it is proprietary or due to complications of working with YouTube data. This method of data release (video ids only) is standard/necessary for YouTube corpora in order to automatically respect video deletions. We understand that these answers may be unsatisfying, but note that several works have already used YouTube-ASL, in addition to the many works that use YouTube or other url-hosted datasets, with minimal data loss over time and benefit to the research community. We expect the same to be true of YouTube-SL-25.

---

> > ### Comment · Reviewer_nLUa · 2024-11-24
> >
> > Thank you for your response and for clarifying. I still believe that the dataset not having a sustainability plan may cause issues in the future with reproducibility of the results proposed in the paper and may affect future publications that use this dataset. However, I understand the value of a large dataset for sign language being extremely beneficial therefore I will increase my score.

---

### Official Review · Reviewer_f36H · 2024-11-03

**Soundness:** 3
**Presentation:** 3
**Contribution:** 3
**Rating:** 6
**Confidence:** 3

**Summary:**

This paper introduces YouTube-SL-25, a large-scale, open-domain, multilingual sign language video corpus with seemingly well-aligned captions. It features over 3,000 hours of videos spanning more than 25 sign languages, making it the largest supervised sign language dataset to date. The corpus surpasses the size of YouTube-ASL, a previous sign language corpus, by more than three times.
Here are the key contributions of the paper:
1. Data Collection and Curation: The paper describes a two-step process for mining and filtering sign language videos from YouTube. First, videos tagged with sign language-related keywords are automatically retrieved. Then, a manual triage process is employed, where a single annotator identifies high-quality channels with well-aligned captions, prioritizing channels based on content duration
2. Multilingual and Open-Domain Focus: Unlike datasets like AfriSign and JWSign, which focus on specific domains (e.g., Bible translations), YouTube-SL-25 includes videos from a diverse range of topics and genres, making it "open-domain".
3. Baseline Models and Multilingual Transfer: The paper establishes baselines for sign language understanding tasks, utilizing a multilingual multitask model based on T5. Results indicate that multilingual transfer learning benefits both resource-rich and resource-poor sign languages within the corpus.
4. Addressing Data Bottleneck and Inclusivity: The paper aims to tackle the scarcity of data for sign language research, particularly for less-studied sign languages. It highlights the importance of developing robust filtering and preprocessing tools for sign language data to further expand dataset size and inclusivity.
5. Ethical Considerations: The authors acknowledge the ethical implications of working with videos featuring individuals and discuss their anonymization efforts using MediaPipe Holistic skeletons. They also emphasize the need for ongoing research to ensure responsible data usage and equitable model performance across different demographics

**Strengths:**

Originality:
The paper introduces YouTube-SL-25, a large-scale, open-domain, multilingual corpus of sign language videos with seemingly well-aligned captions. This is the largest supervised sign language dataset to date and the first or largest parallel dataset for many of its 25+ component languages. YouTube-SL-25 significantly expands upon YouTube-ASL, being over three times its size. The corpus creation methodology, while building upon prior techniques, innovates by utilizing a single annotator for manual triage, prioritizing channels by content duration. This approach allows for efficient curation, enabling the inclusion of a diverse array of sign languages. Additionally, the paper explores the novel integration of sign language identification as a task within translation models.

Quality:
The paper presents a well-constructed and carefully curated dataset. The authors employ a two-step process, first using automatic classifiers to identify potentially relevant videos and then utilizing a manual triage process to ensure quality and alignment. While acknowledging the potential for limitations due to the single-annotator approach, the authors provide detailed descriptions of the annotation standards and conduct audits to assess and report error rates. The dataset's quality is further evidenced by the strong baseline results achieved in translation and language identification tasks.

Clarity:
The paper is well-written and easy to follow. The authors clearly present their methodology, experiments, and findings using concise language and a logical structure. The paper effectively utilizes figures and tables to present data and illustrate key concepts. The authors provide a thorough literature review, contextualizing their work within the existing research landscape and citing relevant prior datasets and models.

Significance:
The paper addresses a critical bottleneck in sign language processing: the scarcity of data, especially for less-studied sign languages. The creation and release of YouTube-SL-25 as an open resource has the potential to significantly advance the field by enabling the development and evaluation of more robust and inclusive sign language processing models. The paper's findings on multilingual transfer learning hold promise for improving the performance of sign language understanding systems across various languages, including those with limited resources. The dataset's size and diversity also make it a valuable asset for exploring novel architectures and training paradigms, potentially leading to breakthroughs in sign language recognition, translation, and generation

**Weaknesses:**

1. Limited Evaluation Scope: While the corpus covers a wide range of sign languages, the baseline experiments presented in the paper only evaluate four sign languages. Expanding the evaluation to encompass a more diverse subset of the languages represented in YouTube-SL-25, especially those with fewer resources, would provide a more comprehensive understanding of the corpus's utility for multilingual transfer learning and the impact of data scale on model performance. This expanded evaluation could reveal specific challenges or opportunities associated with different sign languages, informing future research directions.
2. Single-Annotator Triage: The reliance on a single annotator for manual video triage, while efficient for corpus creation, raises concerns about potential biases and inconsistencies in the data selection process. The annotator's proficiency in certain sign languages and potential blind spots in recognizing subtle cues related to caption alignment or signer proficiency could introduce systematic biases. Involving multiple annotators, ideally with diverse linguistic backgrounds and sign language expertise, would enhance the reliability and objectivity of the triage process. This collaborative approach would help mitigate individual biases and ensure a more balanced representation of signing styles and dialects within each language.
3. Potential Data Noise: The authors acknowledge the potential for noise in the dataset, particularly related to caption quality and alignment. While they employ some filtering mechanisms and conduct audits to assess error rates, the inherent variability in YouTube content production and captioning practices poses a challenge. Developing more robust methods for detecting and filtering out noisy data, potentially leveraging techniques from automatic speech recognition or natural language processing, would improve the overall quality of the corpus. This could involve automated caption alignment algorithms, quality assessment metrics, and possibly even community-based validation initiatives to harness the collective expertise of the Deaf community.
4. Limited Demographic Analysis: The paper provides a basic analysis of signer demographics using automated classifiers for skin tone and perceived gender presentation. However, a more comprehensive and nuanced analysis of demographic representation, considering factors such as age, regional variations, signing styles, and disability intersectionality, is crucial for understanding potential biases and ensuring the inclusivity of models trained on the data. It would also be helpful to discuss the potential implications of these findings for the development of fair and unbiased sign language processing models.
5. Limited Focus on Sign Language Generation: While the paper demonstrates the corpus's usefulness for sign language understanding tasks, it does not address its potential for sign language generation. Developing and evaluating sign language generation models, particularly those capable of producing fluent and expressive signing, remains a challenging area of research. Leveraging YouTube-SL-25 for this purpose could involve exploring techniques such as sequence-to-sequence modeling, conditional variational autoencoders, and reinforcement learning to generate realistic and natural-looking signing.

**Questions:**

Based on the provided excerpts from the paper "YOUTUBE-SL-25: A LARGE-SCALE, OPEN-DOMAIN MULTILINGUAL SIGN LANGUAGE PARALLEL CORPUS" and our previous conversation, here are some questions and suggestions for the authors:
Data Curation and Annotation:
1. Clarify the rationale for the 15-hour threshold for language inclusion. The paper mentions including languages with at least 15 hours of representation, aligning with the minimum size of previous datasets. However, the authors also acknowledge this as "extremely low resource". Further explanation of the trade-offs considered in setting this threshold and how it balances representativeness with data scarcity would be beneficial.
2. Elaborate on the specific challenges encountered during the manual triage process. The paper provides a high-level overview of the triage procedure but could benefit from a more detailed discussion of the practical difficulties faced. For instance, were there specific sign languages that proved particularly challenging to identify or assess for caption alignment? Providing insights into these challenges would shed light on the complexities of working with diverse and low-resource sign language data.
3. Address the potential for bias introduced by the single annotator. While the paper acknowledges this limitation, a more in-depth discussion of mitigation strategies would strengthen the analysis. For example, could the authors elaborate on the annotator's background and expertise in different sign languages? Could they discuss the feasibility of incorporating a second-level review by additional annotators, even for a subset of the data, to assess inter-annotator agreement and identify potential biases?

Dataset Analysis and Evaluation:
1. Expand the demographic analysis to encompass a wider range of factors. The current analysis focuses on skin tone and perceived gender presentation. However, considering other demographic variables, such as age, regional variations in signing, and disability intersectionality, would provide a more comprehensive understanding of the dataset's representativeness.
2. Consider including additional evaluation metrics that capture discourse-level features. The current evaluation focuses primarily on sentence-level translation accuracy. Incorporating metrics that assess the coherence, fluency, and expressiveness of translated output, particularly concerning discourse-level phenomena in sign language, would provide a more nuanced view of model performance.
3. Explore the potential of YouTube-SL-25 for sign language generation tasks. The paper primarily focuses on understanding tasks.
Expanding the scope to include generation tasks could open up new research avenues. Discussing potential challenges and opportunities in this area, and perhaps presenting preliminary experiments on a subset of the data, would enrich the paper's contribution.

Generalizability and Future Work:
1. Discuss the generalizability of the findings to other sign language processing tasks. While the paper focuses on translation and language identification, addressing the dataset's potential for other tasks, such as sign language recognition, summarization, or sentiment analysis, would highlight its broader applicability.
2. Outline future directions for improving the corpus and addressing its limitations. This could include strategies for expanding the dataset to include more languages and signers, refining the annotation process, developing more robust noise filtering techniques, and exploring alternative data sources beyond YouTube.
These questions and suggestions aim to encourage a deeper exploration of the dataset's strengths, limitations, and potential impact. Addressing these points would contribute to a more robust and informative discussion

---

> ### Author Response · Authors · 2024-11-22
>
> Thank you for your valuable comments!
>
> Re Weaknesses:
> 1. On lines 423-424 and 465-469 we explain that the limited selection of benchmarks across sign languages is due to limited availability (including licenses). We highlight this as an opportunity for future work. We believe that our baselines compellingly show the benefit of YouTube-SL-25 for multilingual transfer in high- and low-resource sign languages.
> 2. While more annotators can always be beneficial, the annotations to produce the dataset were relatively coarse and permissive (both in terms of the blanket effect on channels, and the criteria for inclusion in the dataset). The salient potential bias in the annotation procedure, as we acknowledge in Limitations, is that it reduces the number of unique signers because it favors large channels.
> 3. We have shown in experiments that YouTube-SL-25 substantially improves downstream performance, demonstrating the knowledge and signal from YouTube-SL-25 clearly outweighing its noise. We also welcome efforts by the community to further refine YouTube-SL-25 or future datasets in lines 93-98, and acknowledge this as a limitation throughout the paper.
> 4. Additional attributes are always desirable for demographic analysis (especially age could be interesting), but we believe we already met a high standard for the field by predicting skin tone and perceived gender presentation across the entire dataset using classifiers. On lines 491-497 we describe how this dataset transparency still does not guarantee resulting models behave equally for subgroups, and that these factors should receive targeted evaluation.
> 5. Sign language generation is an interesting task, but it is out of scope for this work. Beyond the general immaturity of sign language generation methods, as we mention on lines 459-462 diverse web data poses additional challenges which should be a research topic in themselves. We provide sign language translation and identification baselines that we believe already demonstrate the value of YouTube-SL-25.
>
> Re Questions:
>
> Data Curation and Annotation:
> 1. We explain on lines 82-87 that this threshold is just about the name of the dataset, YouTube-SL-25 instead of YouTube-SL-55, to avoid overstating the functional diversity of the dataset. It does not affect actual inclusion of languages in the dataset, so there are no tradeoffs.
> 2. Specific challenges, heuristics, and observations from the annotation process are described in lines 160-215. For example, in footnote 3 we observe that developing countries tend to have less content or less well-produced content.
> 3. We avoided elaborating too much on the annotator’s sign language background to avoid breaking anonymity. We can provide more detail in the final paper, but overall do not see annotator background or bias as especially significant given the coarseness and permissiveness of the annotations.
>
> Dataset Analysis and Evaluation:
> 1. See Weakness #4.
> 2. Discourse-level features are an interesting topic for future work, but sentence-level metrics still effectively show gaps between our baseline ablations. Note that on 463-464 we acknowledge the limitation that—while our models support discourse-level and other types of tasks—we do not evaluate these capabilities due to lack of uniform benchmarks.
> 3. See Weakness #5.
>
> Generalizability and Future Work:
> 1. On lines 95-97 we state that we expect the dataset to be a useful pretraining resource for downstream sign language understanding tasks generally (including caption alignment), and on 459-462 for sign language generation.
> 2. We describe on lines 97-98 that we expect noisier data could be filtered (using models trained on YouTube-SL-25) to construct the next wave of sign language datasets, on 255-256 that sources of data like Bible translations or national broadcasts could help with language imbalance, and on 465-469 that there is a need for more multilingual sign language benchmarks.

---

### Meta-Review · Area_Chair_21SH · 2024-12-21

**Metareview:**

The article has received evaluations from three reviewers, all of which are positive. The authors have also provided detailed responses to the related issues. Therefore, the article will be accepted.

**Additional Comments On Reviewer Discussion:**

The article has received evaluations from three reviewers, all of which are positive. The authors have also provided detailed responses to the related issues. Therefore, the article will be accepted.

---

### Decision · Program_Chairs · 2025-01-22

Accept (Poster)